# GeoReg: Direct biplanar DSA-to-CTA registration with geodesic consistency for acute ischemic stroke

**Rudolf L. M. van Herten**[*1,2]    RLV4001@MED.CORNELL.EDU
**Robert Graf**[*3,4]    ROBERT.GRAF@TUM.DE
**Felix Bitzer**[3]    FELIX.BITZER@TUM.DE
**Jan S. Kirschke**[3]    JAN.KIRSCHKE@TUM.DE
**Johannes C. Paetzold**[1,2]    JPAETZOLD@MED.CORNELL.EDU

[1] *Department of Radiology, Weill Cornell Medicine, New York, United States of America*

[2] *Cornell Tech, New York, United States of America*

[3] *Department of Neuroradiology, School of Medicine, TUM University Hospital, Technical University of Munich, Munich, Germany*

[4] *Institute for AI and Informatics in Medicine, TUM University Hospital, Technical University of Munich, Munich, Germany*

**Editors:** Accepted for publication at MIDL 2026

## Abstract

The complementary nature of pre-procedural computed tomography angiography (CTA) and intraoperative digital subtraction angiography (DSA) has motivated significant interest in their registration to enhance therapeutic decision-making during stroke interventions. However, current methods depend on accurate vessel segmentation in both modalities, creating a deployment bottleneck due to the requirement for extensive annotated training data. Here, we present an alternative approach that establishes the feasibility of registration without this dependency. Instead of extracting vascular features using pre-trained models, we optimize a direct registration framework using maximum intensity projections of DSA sequences to align a silhouette of the subtracted X-ray image. We introduce a geodesic consistency formulation that jointly optimizes biplanar views, employing soft geometric constraints on SO(3) to maintain consistency while accommodating non-orthogonal scanner configurations. We demonstrate the effectiveness of this model on clinical stroke data and find that it outperforms existing methods, proving particularly effective in escaping local minima where single-view optimization fails. These results indicate that reliable DSA-to-CTA registration is achievable without vessel-specific training data, simplifying the path toward clinical integration.

**Keywords:** Pose estimation, image registration, digital subtraction angiography, computed tomography angiography

## 1. Introduction

Ischemic stroke remains a leading cause of mortality and disability worldwide, affecting over 700,000 people annually in the United States (Powers, 2020), with large vessel occlusions representing the primary mechanism of injury (Malhotra et al., 2017). In recent years,

---

* Contributed equally

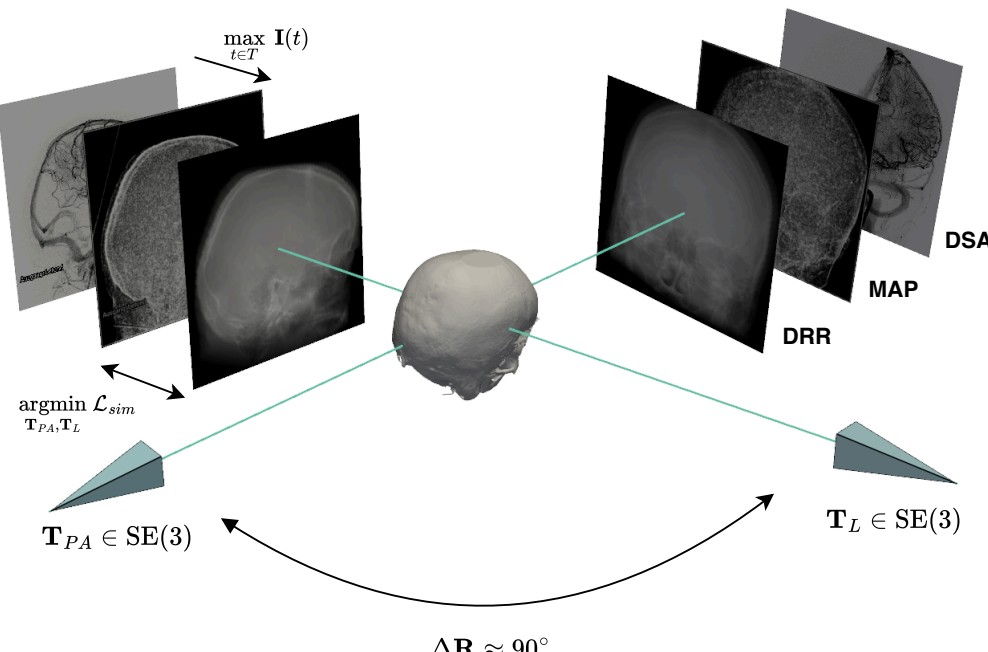

Figure 1: Overview of the proposed optimization setup. The image registration of 2D + t DSA images and 3D CTA image volumes can be formulated as a pose estimation problem. Both posteroanterior (PA) and lateral (L) projection poses $\mathbf{T}_{PA}, \mathbf{T}_L \in \mathrm{SE}(3)$ are simultaneously optimized under a biplanar geometric constraint ($\Delta\mathbf{R} \approx 90°$), minimizing similarity loss $\mathcal{L}_{sim}$ between temporal maximum intensity projections (MAP) silhouettes generated from DSA images and synthetic X-rays (DRR) computed from the CTA volume.

endovascular thrombectomy (EVT) has emerged as the most effective treatment for acute ischemic stroke caused by large vessel occlusion, achieving successful recanalization in over 85% of cases and dramatically improving patient outcomes (Berkhemer et al., 2015). In clinical practice, 3D computed tomography angiography (CTA) is employed for pre-procedural assessment and EVT eligibility determination, while biplanar 2D digital subtraction angiography (DSA) provides intraoperative guidance. While DSA offers superior temporal and spatial resolution with detailed visualization of cerebral perfusion dynamics, it lacks the three-dimensional anatomical context from CTA. Conversely, CTA lacks DSA's temporal information and resolution. The complementary nature of these modalities has motivated significant interest in combining DSA and CTA in image analysis, specifically in DSA-to-CTA registration methods to enhance therapeutic decision-making during interventions.

Previous work on 3D-to-2D registration of cerebrovascular angiography has established several approaches for aligning pre-procedural and intraoperative imaging. Early intensity-based methods (Hipwell et al., 2003) matched digitally reconstructed vasculature from magnetic resonance angiography (MRA) to DSA images, but required both manual segmentation of MRA vasculature and substantial user input to guide the registration process.

Recent advances have explored differentiable rendering for intraoperative 3D-to-2D registration (Gopalakrishnan et al., 2024), with applications to registering vein segmentations between CTA and DSA images (Downs et al., 2025). However, this necessitates accurate segmentation of veins in both modalities to learn proper initializations and perform registration. Biplanar constraints assuming orthogonal posteroanterior (PA) and lateral (L) DSA acquisition angles have been shown to guide registration (Lee et al., 2025). Yet, this relies on fixed heuristics that fail to account for real-world imperfections in scanner positioning and orientation. Notably, all named methods share a dependency on accurate vasculature extraction for registration, requiring high-quality segmentation models trained on extensive annotated datasets. This reliance on vessel-based features represents a bottleneck in clinical deployment, as acquiring matched vascular segmentation across different imaging modalities remains challenging due to inherent differences in resolution, acquisition protocols, and contrast agent dynamics between DSA and CTA.

Here, we present a novel approach that circumvents the dependency on vasculature segmentation and significantly enhances robustness of DSA-to-CTA registration. We achieve this through two key contributions.

1. We propose a direct registration framework that aligns DSA and CTA images without requiring vessel segmentation. By recovering a silhouette of the subtracted initial X-ray using maximum intensity projections (MAP), we enable direct image-similarity-based DSA-to-CTA registration, circumventing the cross-modal correspondence challenges inherent to vessel-based optimization.

2. We introduce a geodesic consistency formulation for biplanar registration that jointly optimizes PA and L views. In contrast to methods that strictly constrain the biplanar geometry (Lee et al., 2025), our approach uses soft penalties to encourage biplanar consistency while allowing flexibility for non-orthogonal scanner configurations.

This direct approach eliminates the need for vessel-specific training data or pre-trained vessel segmentation models, requiring only lightweight skull boundary extraction.

## 2. Preliminaries

DSA is commonly used for pre- and post-intervention assessment in acute ischemic stroke. Through repeated scanning at intervals of $\Delta t = 0.5$ to 2.0s, DSA visualizes contrast agent propagation through the cerebral vasculature, enabling detection of thromboembolic occlusions and assessment of vessel recanalization following mechanical thrombectomy. In a typical intraoperative stroke imaging setting, two DSA sequences are acquired from different angles.

Let $\mathbf{V} : \mathbb{R}^3 \to \mathbb{R}$ denote a 3D CTA image volume and $\mathbf{I} : \mathbb{R}^2 \times \mathbb{T} \to \mathbb{R}$ denote a 2D+t DSA image sequence of $\mathbf{V}$ taken at an unknown camera pose $\mathbf{T} \in \mathrm{SE}(3)$. Specifically, two DSA sequences are acquired such that $\mathbf{I} \in \{\mathbf{I}_{PA}, \mathbf{I}_L\}$. The goal of biplanar DSA-to-CTA registration is to identify the unknown camera poses $\mathbf{T}_{PA}$ and $\mathbf{T}_L$ that align the pre-procedural CTA volume $\mathbf{V}$ with the intraoperative DSA sequences $\mathbf{I}_{PA}$ and $\mathbf{I}_L$, thereby establishing spatial correspondence between the two imaging modalities.

## 2.1. Differentiable X-ray rendering

The alignment of $\mathbf{V}$ and $\mathbf{I}$ requires the extraction of a synthetic X-ray projection $\hat{\mathbf{I}}$ from $\mathbf{V}$ that exists in the same space as $\mathbf{I}$, which allows images to be matched through a similarity metric. This is achieved through a physics-based simulation in which rays $\mathbf{r}$ are cast from a radiation source $\mathbf{s} \in \mathbb{R}^3$ to a target pixel $\mathbf{p} \in \mathbb{R}^3$ on a detector $\mathbf{P} \in \mathbb{R}^{n \times 3}$ such that $\mathbf{r}(\alpha) = \mathbf{s} + \alpha(\mathbf{p} - \mathbf{s})$ with $\alpha \in [0, 1]$. X-ray attenuation values at $\mathbf{p}$ may then be described by the Beer-Lambert law (Swinehart, 1962):

$$\hat{\mathbf{I}}(\mathbf{p}) = \hat{\mathbf{I}}_0 \exp \left( - \| \mathbf{r}'(\alpha) \| \int_0^1 \mathbf{V}(\mathbf{r}(\alpha)) \, d\alpha \right) \tag{1}$$

which defines a line integral of attenuation coefficients along the ray path traversing the volume $\mathbf{V}$ with initial X-ray intensity $\hat{\mathbf{I}}_0$. In practice, this integral is approximated using numerical quadrature with $\mathbf{M}$ sample points as per Siddon (1985):

$$\mathbf{r}'(\alpha) \sum_{m=1}^{\mathbf{M}-1} \mathbf{V} \left[ \mathbf{r} \left( \frac{\alpha_{m+1} + \alpha_m}{2} \right) \right] (\alpha_{m+1} - \alpha_m). \tag{2}$$

When evaluated across all pixels on the detector array $\mathbf{P}$, this process generates a digitally reconstructed radiograph (DRR) $\hat{\mathbf{I}}$. We implement this rendering pipeline using the DiffDRR library (Gopalakrishnan and Golland, 2022), which leverages GPU acceleration and automatic differentiation to achieve real-time performance. The key advantage of this differentiable formulation is that it enables gradient-based optimization of the rigid transformation $\mathbf{T} \in \mathrm{SE}(3)$, allowing for iterative minimization of the discrepancy between synthetic projections $\hat{\mathbf{I}}$ and DSA images $\mathbf{I}$ to achieve accurate 3D-2D registration.

## 2.2. DSA acquisition and processing

A DSA sequence is acquired through temporal subtraction imaging, where initial fluoroscopic images display X-ray absorbing structures as dark regions against a bright background (Kaufman and Lee, 2013). The pre-contrast baseline image $\mathbf{X}_0 : \mathbb{R}^2 \to \mathbb{R}$ captures the native anatomy before contrast administration. In clinical practice, this baseline is inverted to create a mask that nullifies the background anatomical structures. Subsequent frames $\mathbf{X}_t$ are acquired during contrast injection, and the subtraction process isolates the vascular enhancement:

$$\mathbf{I}(t) = \mathbf{X}_t - \mathbf{X}_0, \quad t \in \{1, ..., T\} \tag{3}$$

where $T$ denotes the number of frames captured during the contrast injection phase. This subtraction effectively removes static anatomical structures present in both the mask and contrast frames, leaving only the contrast-enhanced vasculature visible. Unlike unsubtracted angiography where bones obscure vascular structures, DSA produces images where vessels appear as dark structures against a neutral background, significantly improving vessel visibility and enabling detailed assessment of vascular pathology.

To enable registration with DRR projections derived from CTA volumes, we compute the MAP across the temporal dimension:

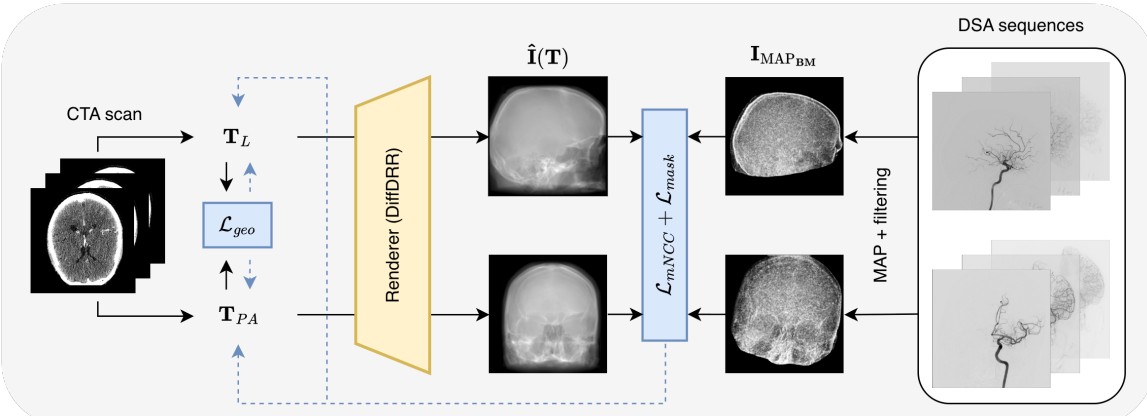

Figure 2: Schematic overview of the proposed method. Posteroanterior and lateral poses $\{\mathbf{T}_{PA}, \mathbf{T}_L\}$ are initialized at a 90° relative angle with respect to the CTA isocenter. DSA sequences are processed using maximum intensity projection (MAP) and filtering to enhance relevant image information in the silhouette ($\mathbf{I}_{\mathrm{MAP}_{\mathrm{BM}}}$). The CTA volume is processed by a DiffDRR renderer (Gopalakrishnan and Golland, 2022) to generate DRRs $\hat{\mathbf{I}}(\mathbf{T})$ that are compared with $\mathbf{I}_{\mathrm{MAP}_{\mathrm{BM}}}$ and optimized using similarity losses $\mathcal{L}_{mNCC} + \mathcal{L}_{mask}$. The geodesic loss $\mathcal{L}_{geo}$ maintains the geometric consistency between views during pose optimization.

$$\mathbf{I}_{\mathrm{MAP}} = \max_{t \in \{1,\dots,T\}} \mathbf{I}(t). \tag{4}$$

While a minimum intensity projection (MIP) would capture all vasculature that appeared throughout the sequence, the MAP produces an inverted representation that shares image similarity with standard DRRs, where dense structures appear bright (see Appendix A). This representation facilitates image registration by aligning similar intensity patterns between the processed DSA and CTA-derived projections (see Figure 1).

## 3. Methods

Our proposed method performs refinement of target images $\mathbf{I}_{\mathrm{MAP}}$ while exploiting biplanar geometric constraints to establish correspondence between CTA volume $\mathbf{V}$ and DSA sequences $\mathbf{I}$. An overview of the proposed method is presented in Figure 2.

### 3.1. Silhouette refinement

Establishing reliable correspondence between $\mathbf{V}$ and $\mathbf{I}$ requires a robust optimization objective. While the MAP representation $\mathbf{I}_{\mathrm{MAP}}$ shares intensity characteristics with DRRs, direct registration faces two critical challenges: (1) significant noise artifacts from extracting suppressed anatomical structures, and (2) field-of-view mismatches between modalities.

To address these challenges, we apply two preprocessing steps. First, we employ bilateral filtering (Wagner et al., 2022) to enhance structural coherence while preserving edges, effec-

tively denoising the narrow dynamic range ($\sim$50 pixel values for a 12-bit detector) inherent to MAP extraction. Second, we automatically extract skull masks using nnU-Net (Isensee et al., 2021) trained on the DSCA dataset (Zhang et al., 2025) to restrict optimization to shared anatomical regions, preventing misalignment with cervical and thoracic structures present only in DSA. The refined target image is obtained as:

$$\mathbf{I}_{\mathrm{MAP_{BM}}} = \mathcal{F}_B(\mathbf{I}_{\mathrm{MAP}}) \odot \mathcal{M} \tag{5}$$

where $\mathcal{F}_B$ denotes the bilateral filtering operation, $\mathcal{M}$ represents the binary skull mask, and $\odot$ denotes element-wise multiplication. This preprocessing creates a robust optimization objective by suppressing noise while constraining registration to anatomically corresponding regions. Implementation details are provided in Appendix B.1.

### 3.2. Optimization criterion

Given the refined reference images $\mathbf{I}_{\mathrm{MAP}_{BM}}$ and masks $\mathcal{M}$ derived in Section 3.1, we formulate the registration as a joint optimization over the biplanar pose parameters. Our objective function combines image similarity metrics with geometric consistency constraints to simultaneously optimize both posteroanterior and lateral acquisition poses.

**Single-view optimization:** For each view independently, we seek the pose $\hat{\mathbf{T}} \in \mathrm{SE}(3)$ that best aligns the processed DSA image with the CTA-derived projection:

$$\hat{\mathbf{T}} = \arg\min_{\mathbf{T}} \left[ \alpha \mathcal{L}_{\mathrm{mNCC}}(\mathbf{T}) + (1 - \alpha)\mathcal{L}_{\mathrm{mask}}(\mathbf{T}) \right] \tag{6}$$

The primary term $\mathcal{L}_{\mathrm{mNCC}}$ measures image similarity via the multiscale normalized cross-correlation (Gopalakrishnan et al., 2024) between the filtered MAP image $\mathbf{I}_{\mathrm{MAP}_{BM}}$ and the DRR $\hat{\mathbf{I}}(\mathbf{T})$ rendered at pose $\mathbf{T}$. The secondary term $\mathcal{L}_{\mathrm{mask}}$ enforces spatial consistency through a generalized dice loss (Sudre et al., 2017) between sigmoid-transformed DRR intensities $\sigma(\hat{\mathbf{I}}(\mathbf{T}))$ and the cranium mask $\mathcal{M}$. The parameter $\alpha \in [0, 1]$ balances image similarity against spatial overlap.

**Biplanar optimization:** Independent optimization of PA and L views may converge to locally optimal solutions without exploiting their known geometric relationship. We therefore jointly optimize both poses while maintaining approximate orthogonality:

$$\{\hat{\mathbf{T}}_{\mathrm{PA}}, \hat{\mathbf{T}}_{\mathrm{L}}\} = \arg\min_{\mathbf{T}_{\mathrm{PA}}, \mathbf{T}_{\mathrm{L}}} \left[ \beta \mathcal{L}_{\mathbf{T}_{\mathrm{PA}}} + (2 - \beta)\mathcal{L}_{\mathbf{T}_{\mathrm{L}}} + \lambda \mathcal{L}_{\mathrm{geo}} \right] \tag{7}$$

where $\mathcal{L}_{\mathbf{T}_{\mathrm{PA}}}$ and $\mathcal{L}_{\mathbf{T}_{\mathrm{L}}}$ denote the respective single-view losses (see Equation (6)), with $\beta \in [0, 2]$ allowing asymmetric weighting.

The geodesic consistency term maintains approximate orthogonality between acquisition angles by penalizing deviations from 90° separation:

$$\mathcal{L}_{\mathrm{geo}} = \left| \|\log(\mathbf{R}_{\mathrm{PA}}^{-1}\mathbf{R}_{\mathrm{L}})\| - \frac{\pi}{2} \right| \tag{8}$$

where $\mathbf{R}_{\mathrm{PA}}, \mathbf{R}_{\mathrm{L}} \in \mathrm{SO}(3)$ are the rotation components of the respective poses. The matrix logarithm $\log(\cdot)$ maps the relative rotation $\mathbf{R}_{\mathrm{PA}}^{-1}\mathbf{R}_{\mathrm{L}}$ from the Lie group $\mathrm{SO}(3)$ to its Lie

algebra $\mathfrak{so}(3)$, where the norm yields the geodesic angle between views. This soft constraint encourages approximately orthogonal acquisition angles while allowing flexibility for non-ideal scanner configurations, where the weighting parameter $\lambda$ controls the strength of the geometric constraint relative to image-based alignment objectives.

Details on hyperparameter settings are provided in Appendix B.2, and code is made publicly available on GitHub.

## 4. Experiments

To assess direct DSA-to-CTA registration, we test on a clinical stroke dataset with comparisons to existing optimization-based approaches.

### 4.1. Data

All experiments in this work are conducted on a subset of the ISLES'24 challenge dataset (de la Rosa et al., 2024). We utilize the publicly available CTA volumes from ISLES'24, which were acquired as part of standard stroke assessment protocols. A subset of these volumes was matched with an internal collection of biplanar DSA acquisitions obtained during endovascular procedures at the TUM University Hospital. The DSA sequences consist of paired PA and L projections acquired using standard neurovascular imaging protocols. Our experimental cohort comprises 77 patients for whom both pre-procedural CTA and intraoperative biplanar DSA imaging were available, enabling paired evaluation of registration.

### 4.2. Evaluation

Since our method does not require vessel segmentations for registration, we can leverage these structures as an independent validation of registration accuracy. Our evaluation strategy measures the alignment quality by comparing vessel structures between the registered DRRs and original DSA images, focusing on two key anatomical regions: (1) the carotid arteries and (2) the venous sinuses.

**Vessel segmentations:** To obtain vessel segmentations across different imaging modalities, we leveraged multiple specialized methods. For CTA images, we trained two nnU-Net models (Isensee et al., 2021) targeting different vascular structures. For carotid artery segmentation, we utilized an in-house dataset of 171 CTA scans, where labels were generated using a publicly available prediction network (Tiefenthaler et al., 2025) and subsequently filtered for errors. For venous sinus segmentation, we trained on the CTA subset ($N = 25$) of the TopBrain challenge dataset (Yang et al., 2025). For DSA images, we similarly employed complementary approaches for arterial and venous vessels. Carotid arteries were segmented using nnU-Net trained on the publicly available DSCA dataset ($N = 180$) (Zhang et al., 2025), which processes DSA MIP images. Venous structures were extracted from DSA image sequences using the CAVE segmentation algorithm (Su et al., 2024).

The segmentation algorithms were applied to all available paired CTA-DSA data. To ensure reliable validation metrics, we excluded any cases where segmentation failed for either modality, as incomplete vessel extraction would invalidate the registration accuracy

Table 1: Quantitative results of DSA-to-CTA registration methods for 59 patients from the ISLES'24 dataset. Performance is assessed using Dice coefficient, clDice for topological consistency, and mean centerline distance ($MCD_p$) for carotid arteries (CA) and venous sinuses (VS) in both posteroanterior (PA) and lateral (L) views. The first row (-) indicates overlap and errors at the initial rendering position. We compare against vanilla DiffDRR, randomized smoothing (RS), and orthogonally-constrained optimization. Best results are shown in bold, and results are given as mean (standard deviation), with statistical significance indicated by asterisks (*).

| | Method | Dice ↑ | | clDice ↑ | | $MCD_p \downarrow$ (mm) | |
|---|---|---|---|---|---|---|---|
| | | CA | VS | CA | VS | CA | VS |
| PA | - | 0.12 (0.10) | 0.30 (0.17) | 0.12 (0.11) | 0.30 (0.15) | 16.98 (13.73) | 8.33 (6.25) |
| | DiffDRR** | 0.32 (0.20) | 0.44 (0.19) | 0.36 (0.29) | 0.45 (0.19) | 7.92 (6.59) | 5.37 (3.01) |
| | RS*** | 0.34 (0.25) | 0.47 (0.20) | 0.37 (0.29) | 0.47 (0.20) | 7.76 (5.88) | 5.60 (3.45) |
| | Orthogonal*** | 0.29 (0.27) | 0.42 (0.21) | 0.32 (0.30) | 0.40 (0.22) | 10.58 (9.14) | 6.43 (4.39) |
| | Ours (single) | 0.44 (0.25) | 0.48 (0.20) | 0.49 (0.29) | 0.48 (0.21) | 6.10 (5.49) | **5.22** (3.08) |
| | Ours (biplanar) | **0.46** (0.26) | **0.49** (0.22) | **0.51** (0.31) | **0.50** (0.22) | **5.74** (5.59) | 5.58 (3.55) |
| L | - | 0.06 (0.12) | 0.18 (0.06) | 0.07 (0.14) | 0.19 (0.06) | 36.52 (22.93) | 6.34 (2.54) |
| | DiffDRR*** | 0.32 (0.20) | 0.42 (0.14) | 0.34 (0.23) | 0.45 (0.18) | 7.67 (5.03) | 3.51 (1.36) |
| | RS** | 0.32 (0.23) | 0.42 (0.14) | 0.35 (0.27) | 0.46 (0.18) | 8.08 (6.30) | 3.20 (1.22) |
| | Orthogonal*** | 0.29 (0.20) | 0.38 (0.13) | 0.30 (0.22) | 0.40 (0.16) | 7.87 (4.83) | 3.75 (1.34) |
| | Ours (single) | 0.41 (0.22) | 0.42 (0.13) | 0.45 (0.26) | 0.45 (0.17) | 6.22 (5.44) | 3.56 (1.38) |
| | Ours (biplanar) | **0.44** (0.22) | **0.43** (0.14) | **0.48** (0.25) | **0.46** (0.18) | **5.86** (5.25) | **3.41** (1.26) |

Asterisks indicate statistical significance compared to our biplanar method using the Wilcoxon signed-rank test for the $MCD_p$ of CA: * $p < 0.05$, ** $p < 0.01$, *** $p < 0.001$.

assessment. This resulted in a subset of 59 patients with successful vessel segmentation across both imaging modalities for evaluation.

**Evaluation metrics:** We evaluate registration accuracy on the detector grid using multiple complementary metrics. For centerline accuracy, we compute the unidirectional mean projected centerline distance ($MCD_p$) from CTA to DSA segmentations:

$$MCD_p = \frac{1}{|P|} \sum_{p \in P} \min_{r \in R} ||p - r|| \tag{9}$$

where $P$ and $R$ represent the projected and reference vessel centerlines, respectively. We use the unidirectional formulation for $MCD_p$ as the superior resolution of DSA images typically yields more complete vessel representations that may not be fully captured in CTA due to resolution limitations. Additionally, we compute the standard Dice coefficient for volumetric overlap and clDice (Shit et al., 2021) for topological consistency (Berger et al., 2025), both evaluated bidirectionally between the projected CTA and DSA vessel masks. To generate DRR vessel masks for comparison, we project the segmented CTA vasculature using the estimated registration pose.

**Baseline methods:** We compare the proposed method against three baseline methods

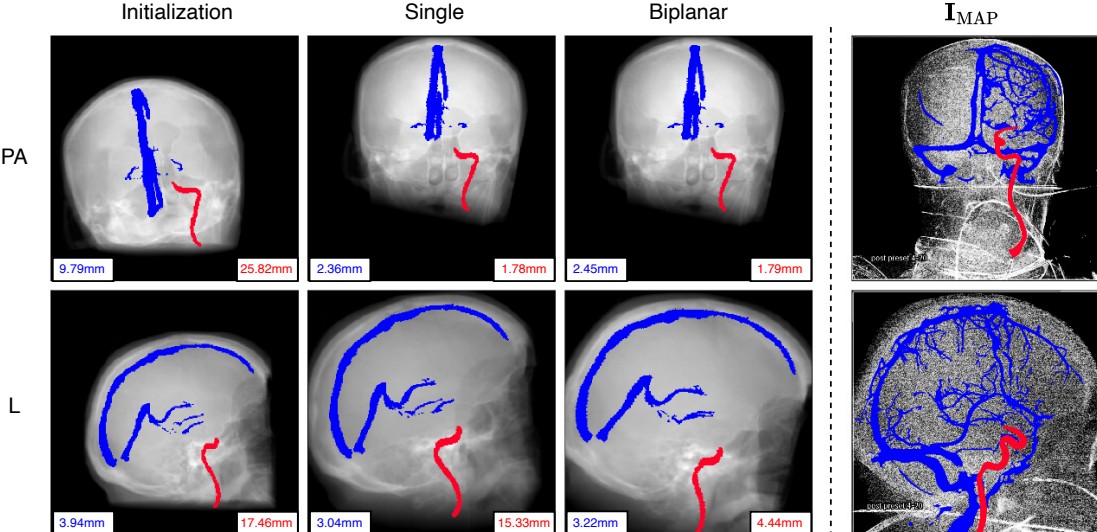

Figure 3: Qualitative example of pose estimation showing registration errors for initialization, single-view, and biplanar optimization. Venous sinus (blue) and carotid artery (red) segmentations are given for both posteroanterior (PA) and lateral (L) views, with mean centerline distance ($\mathrm{MCD}_p$) to DSA segmentations indicated. The rightmost column shows the DSA maximum intensity projection ($\mathbf{I}_{\mathrm{MAP}}$) for reference. Note how our biplanar optimization allows the pose estimator to escape a local minimum and identify the correct rotation in the lateral view.

for optimization-based DSA-to-CTA registration. First, we compare against DiffDRR optimization with default settings, where PA and L poses are optimized independently using gradient descent on the image similarity $\mathcal{L}_{mNCC}$ (Gopalakrishnan and Golland, 2022). Second, we compare against randomized gradient smoothing with adaptive stochastic gradient descent (Sun et al., 2017), which applies Gaussian noise to transformation parameters during optimization to smooth the cost function landscape and escape local minima. Finally, we compare against the strictly orthogonal method proposed in Lee et al. (2025), which enforces perpendicular biplanar geometry using Rodrigues' rotation formula to couple PA and L views with a hard constraint assuming exactly 90° between acquisition angles. Implementation details of baseline methods are provided in Appendix C.

## 5. Results

Table 1 presents the average quantitative registration accuracy for 59 patients from the ISLES'24 dataset. All baseline methods show similar performance, with modest improvements over the initial unregistered position. The strictly orthogonal constraint method performs slightly worse than unconstrained optimization, suggesting that hard geometric constraints may be overly restrictive given the inherent deviations from perfect orthogonality in clinical biplanar acquisitions.

Table 2: Ablations on components included in the final proposed method. $\mathcal{F}_B$ is the bilateral filter, $\alpha$ is the weighting factor for the cranium dice loss, and $\lambda$ is the geodesic consistency weighting factor.

|  | $\mathcal{F}_B$ | $\alpha$ | $\lambda$ | Dice ↑ | | clDice ↑ | | MCD$_p$ ↓ (mm) | |
|---|---|---|---|---|---|---|---|---|---|
|  |  |  |  | CA | VS | CA | VS | CA | VS |
| PA |  | 0 | - | 0.32 (0.20) | 0.44 (0.19) | 0.36 (0.29) | 0.45 (0.19) | 7.92 (6.59) | 5.37 (3.01) |
|  | ✓ | 0 | - | 0.40 (0.25) | 0.49 (0.19) | 0.43 (0.29) | 0.49 (0.19) | 6.51 (5.44) | **5.05** (2.80) |
|  | ✓ | 0.5 | - | 0.44 (0.25) | 0.48 (0.20) | 0.49 (0.29) | 0.48 (0.21) | 6.10 (5.49) | 5.22 (3.08) |
|  | ✓ | 0.5 | 0.1 | **0.46** (0.26) | **0.49** (0.22) | **0.51** (0.31) | **0.50** (0.22) | **5.74** (5.59) | 5.59 (3.55) |
|  | ✓ | 0.5 | 0.5 | 0.40 (0.23) | 0.46 (0.21) | 0.43 (0.27) | 0.47 (0.20) | 6.23 (5.15) | 5.89 (3.43) |
| L |  | 0 | - | 0.32 (0.20) | 0.42 (0.14) | 0.34 (0.23) | 0.45 (0.18) | 7.67 (5.03) | 3.51 (1.36) |
|  | ✓ | 0 | - | 0.40 (0.22) | 0.40 (0.14) | 0.43 (0.25) | 0.44 (0.18) | 6.07 (4.69) | 3.44 (1.32) |
|  | ✓ | 0.5 | - | 0.41 (0.22) | 0.42 (0.13) | 0.45 (0.26) | 0.45 (0.17) | 6.22 (5.44) | 3.56 (1.38) |
|  | ✓ | 0.5 | 0.1 | **0.44** (0.22) | **0.43** (0.14) | **0.48** (0.25) | **0.46** (0.18) | **5.86** (5.25) | **3.41** (1.26) |
|  | ✓ | 0.5 | 0.5 | 0.37 (0.20) | 0.42 (0.13) | 0.40 (0.24) | 0.45 (0.16) | 7.07 (5.64) | 3.53 (1.26) |

Our single-view method outperforms all baselines across all metrics, while the biplanar variant with geodesic consistency achieves the best overall performance, reducing mean centerline distances for carotid arteries by approximately 25% compared to the best baseline method. These improvements are consistent across both PA and L views, demonstrating that soft geometric constraints effectively exploit the biplanar relationship while maintaining flexibility for non-ideal scanner configurations. A qualitative comparison demonstrating successful pose recovery is presented in Figure 3. To characterize failure modes, we analyzed all cases where MCD$_p$ ¿ 10mm for carotid arteries (N=8). These outliers fall into three categories: non-standard head positioning causing poor initialization (N=5), out-of-frame CTA acquisition resulting in incomplete skull rendering (N=2), and low DSA image quality yielding insufficient structural information (N=1). A representative example of each failure mode is shown in Figure 4.

An ablation on loss components is provided in Table 2. For geodesic consistency, $\lambda = 0.1$ was empirically found to perform well, as higher lambdas resulted in performance degradation. This finding aligns with our results on strictly orthogonal rotation, which also resulted in lower performance (see Table 1). All other hyperparameters are as described in Appendix B.2.

Total registration time was ~70s per case (0.18 s/iteration) for both single and biplanar optimization on an NVIDIA L40S GPU, or 35s (0.09 s/iteration) on an NVIDIA H100 GPU.

## 6. Discussion and Conclusion

We have presented a direct approach to biplanar DSA-to-CTA registration that circumvents vessel segmentation through MAP-based silhouette extraction and geodesic consistency constraints. Our experiments demonstrate that reliable registration is achievable without the computational overhead of vessel-specific segmentation models, establishing the feasibility of silhouette-based registration for acute ischemic stroke imaging.

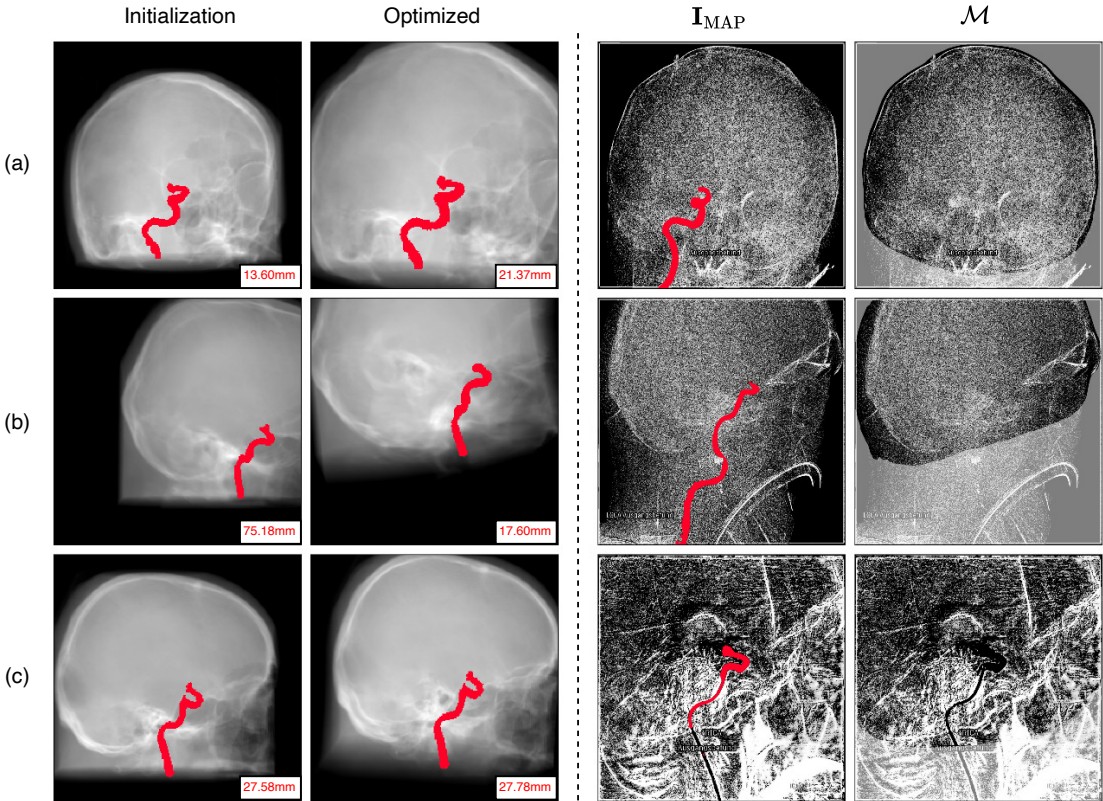

Figure 4: Failure mode analysis for cases with carotid artery $MCD_p$ ¿ 10mm. Left columns show DRR projections at initialization and after optimization with carotid artery segmentations overlaid (red) and $MCD_p$ indicated. Right columns show the corresponding DSA maximum intensity projection $I_{MAP}$ and their corresponding masks highlighted in $\mathcal{M}$. (a) Non-standard head positioning: the patient's head is heavily rotated relative to the standard DSA acquisition angle, causing convergence to a local minimum. (b) Out-of-frame CTA acquisition results in partial skull rendering, degrading the optimization objective. (c) Heavy artifacts in $I_{MAP}$ produce insufficient structural information for robust registration.

The primary performance gains stem from bilateral filtering and the cranium overlap loss, which together enable robust optimization despite noisy MAP representations. The geodesic consistency term provides additional benefit in challenging cases prone to local minima convergence, as illustrated in Figure 3. While our method requires skull segmentation from DSA, this is a lightweight task as cranial boundaries are high-contrast structures readily identifiable even in low-quality MAP images. Segmentation succeeded in all 77 cases, requiring minimal annotation effort (6 hours for 116 training images) with negligible inference overhead (0.1s per image). We expect that other mechanisms for restricting the optimization to the relevant anatomical region would be similarly effective.

While our evaluation is limited to single institution data, the optimization-based nature of our approach may confer robustness to domain shifts. Intensity matching criteria generalize across scanner configurations, and pairwise registration inherently accommodates patient-specific anatomical variation. Preliminary evidence supporting this is provided by successful registration of an external case from Radiopaedia (see Appendix D). Nevertheless, comprehensive validation on multi-center data remains important future work.

Several optimization strategies could further reduce current runtime: sparse implementations of normalized cross-correlation would decrease computational overhead (Gopalakrishnan et al., 2024), CUDA kernel implementations of the forward projection could replace the current PyTorch-based approach (Jiang et al., 2025), while improved hyperparameter selection could improve convergence rates. Nevertheless, sensitivity to initialization represents a fundamental barrier to clinical translation. We found that non-standard patient positioning (e.g., a sagging head) in either modality leads to increased registration errors (see Figure 4) - a critical consideration given the time-sensitive nature of stroke intervention.

While our optimization-based approach demonstrates feasibility, a learning-based pipeline could enable real-time clinical deployment. Recent work explores learned initialization through patient-specific pretraining (Gopalakrishnan et al., 2024), but this paradigm is incompatible with acute stroke care where treatment delays impact outcomes.

We envision training a general pose estimation network on skull anatomy alone, mapping DSA projections directly to CTA poses without patient-specific pretraining. This approach could leverage synthetic projections from a template CTA, learning the mapping between skull silhouettes and poses. Domain-adversarial training with gradient reversal (Ganin and Lempitsky, 2015) could further bridge the appearance gap between modalities, learning pose-discriminative features that are invariant to whether the input is a DSA projection or CTA-derived DRR. Recent work by Downs et al. (2025) demonstrates the feasibility of learning-based initialization through venous segmentations, establishing that generalizable pose estimation is achievable. Our results suggest that vessel-specific features may not be necessary, as skull boundaries alone provide strong geometric constraints for accurate registration. This offers a complementary approach that simplifies training requirements and broadens applicability to arterial-phase acquisitions, where venous structures are unavailable, and the superior resolution of DSA creates correspondence ambiguities with CTA vasculature. Implementation would require standardizing CTA poses across patients, as variations in head positioning make unconstrained pose estimation ill-posed. Alternative approaches merit exploration, including generative models for direct DSA-to-DRR translation (Isola et al., 2017; van Herten et al., 2024) and NeRF-based reconstruction from biplanar projections (Maas et al., 2025; Frisken et al., 2025).

In conclusion, we demonstrate that vessel segmentation-free registration of DSA and CTA is feasible and outperforms existing optimization strategies when combined with appropriate silhouette preprocessing and geometric constraints. By circumventing the need for annotated vascular data, our work paves the way for future learning-based systems capable of providing instant spatial correspondence in time-critical stroke interventions.

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

## Appendix A. DSA derivatives

For visual reference, we present the variations of DSA derivatives in Figure 5. DSA imaging effectively filters out any image intensities not subject to contrast enhancement, resulting in a subtraction of rigid bodies. Inherent acquisition noise enables us to recover a silhouette of the subtracted X-ray image.

## Appendix B. Implementation details

### B.1. Preprocessing

**Bilateral filtering:** Bilateral filtering is implemented through a GPU-accelerated CUDA library (Wagner et al., 2022) that follows standard bilateral smoothing:

$$\mathbf{I}_{\mathrm{MAP}_B}(\mathbf{p}) = \frac{1}{W} \sum_{\mathbf{q}\in\Omega} G_{\sigma_s}(\|\mathbf{p} - \mathbf{q}\|) G_{\sigma_r}(|\mathbf{I}_{\mathrm{MAP}}(\mathbf{p}) - \mathbf{I}_{\mathrm{MAP}}(\mathbf{q})|) \mathbf{I}_{\mathrm{MAP}}(\mathbf{q}) \tag{10}$$

where $G_{\sigma_s}$ and $G_{\sigma_r}$ denote Gaussian kernels for spatial and intensity-based weighting respectively, $\Omega$ represents the local neighborhood of pixel $\mathbf{p}$, and $W = \sum_{\mathbf{q}\in\Omega} G_{\sigma_s}(\|\mathbf{p} - \mathbf{q}\|) G_{\sigma_r}(|\mathbf{I}_{\mathrm{MAP}}(\mathbf{p}) - \mathbf{I}_{\mathrm{MAP}}(\mathbf{q})|)$ ensures normalization of intensity values. This filtering preserves skull boundaries and major anatomical landmarks while suppressing acquisition noise.

**Cranium segmentation:** A simple 2D nnU-Net (Isensee et al., 2021) was trained for

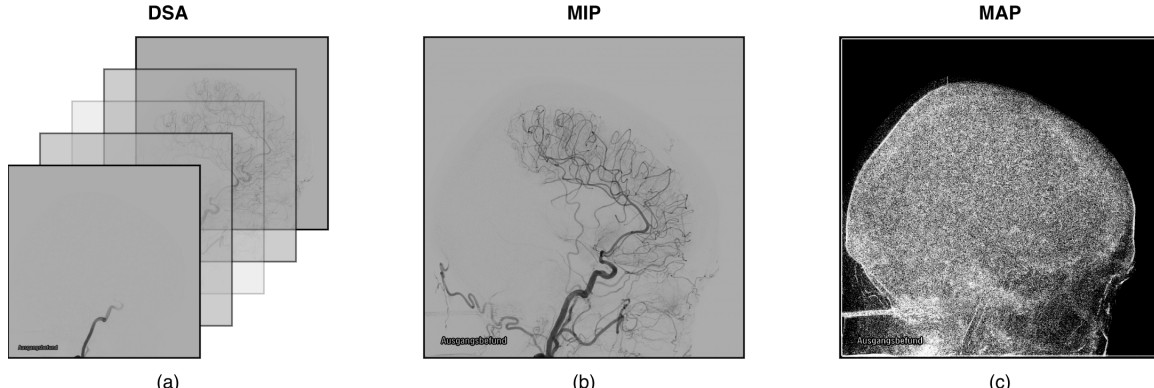

Figure 5: DSA imaging and its derivatives, with (a) the original full temporal DSA sequence, (b) the minimum intensity projection across the time dimension, highlighting all cerebral perfusion, and (c) the maximum intensity projection, highlighting the subtracted X-ray through compounded noise across temporal frames.

cranium segmentation in DSA images. A single model was trained using 116 MAP images (58 PA) from the DSCA dataset with FOLD=ALL configuration. Annotations were performed by an expert with 4 years of experience in medical imaging, with a total annotation burden of approximately 6 hours. The model was subsequently evaluated on the internal ISLES'24 DSA dataset with an average processing time of 0.1 seconds per image on an NVIDIA L40S GPU. Visual inspection confirmed robust segmentation across all cases, underscoring the simplicity of the task and reliable performance with minimal overhead.

### B.2. Optimization settings

Table 3 presents all hyperparameter settings for the proposed biplanar method. The single-view variant employs the exact same hyperparameters, except we optimize for Equation 6 instead (thus ignoring $\beta$ and $\lambda$). For the purpose of computational feasibility, DSA images are reshaped to $256 \times 256$ grids prior to registration, allowing for comparison with DRRs. During optimization, rotation updates are expressed in radians, while translation updates are expressed in decimeters to achieve similar magnitude gradient updates. A OneCycleLR scheduler further prevented overshooting the optimal pose during the initial optimization steps, while ensuring smooth convergence near the end of optimization.

## Appendix C. Baseline implementation

We provide implementation details for the three baseline methods used in our comparison. All baseline methods utilize the cranium mask $\mathcal{M}$ to constrain the optimization region, even when $\alpha = 0$.

**Vanilla DiffDRR:** The vanilla DiffDRR baseline follows the same optimization pipeline as our single-view method with two key modifications:

Table 3: Implementation details for biplanar optimization.

|  | config | value |
|---|---|---|
| $\mathcal{F}_B$ | $\sigma_s$ | 11 |
|  | $\sigma_r$ | 11 |
| Renderer | sdd | from metadata |
|  | sid | (sdd - 400) mm |
|  | size | $[256 \times 256]$ |
|  | spacing | $[1.2 \times 1.2]$ mm$^2$ |
|  | $\mathbf{R}_{PA,0}$ | $[0, 0, 0]$ ° |
|  | $\mathbf{R}_{L,0}$ | $[90, 0, 0]$ ° |
|  | $\mathbf{t}_{[PA,L],0}$ | $[0, \text{sid}, 0]$ mm |
| Optimization | $\mathcal{L}_{mNCC}$ | $s_p = [\text{None}, 13]$ $w_p = [0.5, 0.5]$ |
|  | $\alpha$ | 0.5 |
|  | $\beta$ | 1 |
|  | $\lambda$ | 0.1 |
|  | optimizer | AdamW |
|  | optimizer momentum | $\beta_1, \beta_2 = [0.9, 0.999]$ |
|  | base learning rate | 1e-4 |
|  | weight decay | 0 |
|  | learning rate schedule | OneCycleLR |
|  | max learning rate | 1e-2 |
|  | iterations | 400 |

- No bilateral filtering is applied ($\mathbf{I}_{\text{MAP}_{\text{BM}}} = \mathbf{I}_{\text{MAP}} \odot \mathcal{M}$)

- The mask loss weight is set to $\alpha = 0$, optimizing only $\mathcal{L}_{\text{mNCC}}$

All other hyperparameters remain identical to those specified in Table 3 for single-view optimization.

**Randomized Smoothing (RS):** Following Sun et al. (2017), we implement randomized gradient smoothing using adaptive stochastic gradient descent (ASGD). The optimizer configuration and stochastic perturbation parameters are:

All other settings (preprocessing, renderer configuration) remain identical to vanilla Diff-DRR.

**Orthogonal Constraint:** The orthogonally-constrained baseline enforces strict perpendicular geometry between the views. We formulate the optimization using the PA view as the base parameterization. We compute the orthogonal transition matrix $\mathbf{W}$ using Rodrigues' rotation formula, defined by a fixed 90° angle and a rotation axis $\mathbf{a}$ extracted from the PA rotation ($\mathbf{a} = \mathbf{R}_{\text{PA}}[:, 1]$). The corresponding orthogonal view rotation is then computed as:

Table 4: Randomized smoothing hyperparameters

|  | config | value |
|---|---|---|
| ASGD optimizer | $a$ (power) | 1 |
|  | $A$ (stability constant) | 20 |
|  | $f_{\min}$ (lower bound) | -0.5 |
|  | $f_{\max}$ (upper bound) | 1.0 |
|  | $\omega$ (averaging factor) | 1.0 |
| Gradient smoothing | $Q$ (perturbations) | 6 |
|  | $\sigma_{\mathrm{rot}}$ (rotation noise) | 5 $^{\circ}$ |
|  | $\sigma_{\mathrm{tra}}$ (translation noise) | 5 mm |
|  | $\lambda$ (decay factor) | 0.15 |

$$\mathbf{R}_{\mathrm{L}} = \mathbf{W}\mathbf{R}_{\mathrm{PA}} \tag{11}$$

Unlike the original implementation by Lee et al. (2025) which optimizes 6 parameters (single view only), we found that constraining translations led to convergence issues. Therefore, we optimize translations independently for each view while maintaining the rotation constraint, resulting in 9 total parameters. All other hyperparameters follow Table 3, with the geometric constraint enforced through the Rodrigues formula rather than the soft geodesic penalty.

## Appendix D. External case evaluation

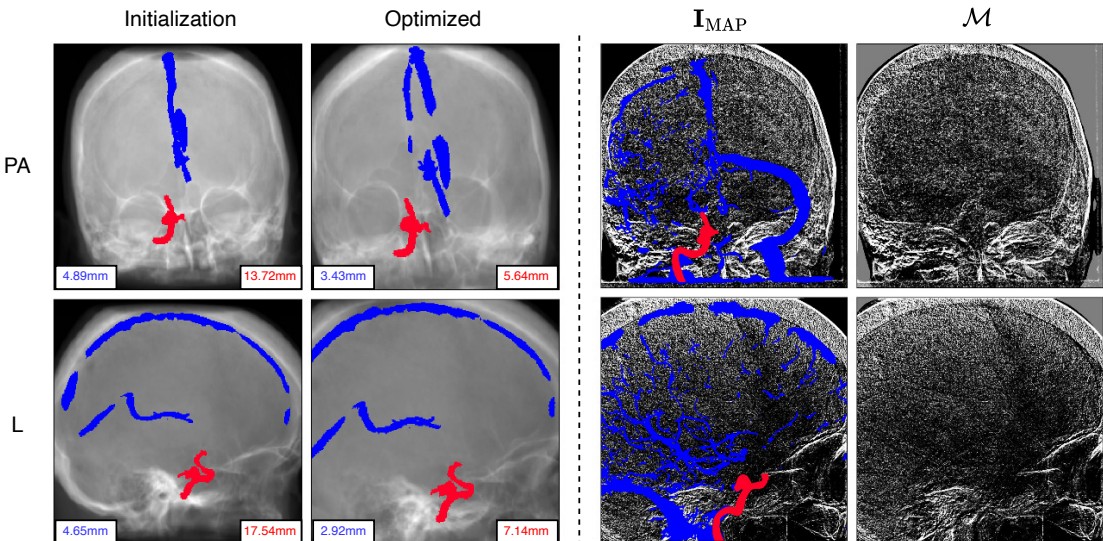

Figure 6: Qualitative evaluation on an external case obtained from Radiopaedia (McKinney, 2024). Left columns show DRR projections at initialization and after optimization with venous sinus (blue) and carotid artery (red) segmentations overlaid and $MCD_p$ indicated. Right columns show the corresponding DSA maximum intensity projection $I_{MAP}$ and cranium mask $\mathcal{M}$. Despite differences in scanner configuration and acquisition protocol, our method successfully registers both views.

