# OpenReview forum: "GeoReg: Direct biplanar DSA-to-CTA registration with geodesic consistency for acute ischemic stroke"
_MIDL.io/2026/Conference — MIDL 2026 Poster_

### Official Review · Reviewer_nLRR · 2025-12-28

**Confidence:** 3
**Preliminary Rating:** 4
**Final Rating:** 4

**Summary:**

The authors propose a direct biplanar registration method (GeoReg) between intraoperative DSA and pre-procedural 3D CTA for acute ischemic stroke, to avoid reliance on vessel segmentation during optimization. The proposed approach computes a temporal maximum-intensity projection of the DSA sequence and refines it via bilateral filtering and skull masking to form a silhouette-like target for matching against differentiably rendered DRRs from CTA (DiffDRR). The core contribution is a soft biplanar geodesic consistency loss on SO(3) that encourages separation between PA and lateral views while allowing non-ideal scanner configurations, this enables joint pose optimization in SE(3).  On a paired clinical cohort derived from ISLES’24 and internal biplanar DSA, the biplanar variant improves Dice/clDice and reduces MCDp compared to baselines.

**Strengths:**

- The paper targets a clinically meaningful and less explored problem of CTA-DSA registration.
- And the proposed segmentation-free registration framing is well-motivated.
- Evaluation uses multiple metrics (Dice, clDice, MCDp) and compares against several reasonable optimization baselines; the qualitative example further illustrates a plausible failure mode (local minimum in single-view) that biplanar coupling can fix.
- The overall presentation is clear and well-organized.

**Weaknesses:**

- The central claim of registration without vessel segmentation dependency is somewhat overstated because the method still depends on a DSA skull mask produced by a trained network,
- Methodologically, the objective mixes mNCC with a skull overlap term, the strongest gains appear to come from preprocessing + mask loss --- while the geodesic term provides modest average improvements and can degrade if $\lambda$ increases, suggesting sensitivity and limited robustness of the geometry prior.
- The authors mentioned that head pose deviations can increase errors, yet no detailed analysis is further provided.

**Detailed Comments:**

- Add clarification on the “segmentation-free” claim: please rephrase to “no vessel segmentation required for optimization,” since skull segmentation is still required and is learned. Quantify how often skull segmentation fails and whether failures correlate with poor registration.

**Justification Of Final Rating:**

Thank you for your response and the additional experimental results and discussions.

My minor concerns have been addressed. I believe this work will be of general interest to the MIDL audience, and I will keep my acceptance recommendation.

**Justification Of The Preliminary Rating:**

The paper addresses an important clinical registration problem and introduces a clean, principled addition that uses soft geodesic biplanar coupling on top of differentiable DRR optimization, with encouraging improvements over several optimization baselines. However, some concerns and questions remain, see the above section.  Therefore, I recommend a weak accept.

**Questions To Address In The Rebuttal:**

- Quantify how often skull segmentation fails and whether these failures correlate with poor registration.
- How sensitive are results to initialization (pose priors) and to hyperparameters?
- Is there a practical solution to reduce runtime from around 4min to clinically viable latency (e.g., less than 1min) without requiring patient-specific pretraining?

---

> ### Author Response · Authors · 2026-01-23
> **Response to Reviewer nLRR (Part 1)**
>
> We appreciate the Reviewer's insightful comments, which we address below.
>
> **Weaknesses:**
>
> **(W1)** The central claim of registration without vessel segmentation dependency is somewhat overstated because the method still depends on a DSA skull mask produced by a trained network.
>
> **(A1)** While our method does indeed require skull segmentation, we note that this is substantially simpler than vessel segmentation: cranial boundaries are high-contrast, geometrically simple structures that required only 6 hours of annotation for 116 images (**Appendix B.1**), compared to the extensive effort needed for cross-modal vessel correspondence. We agree that this should be clarified, which we now do explicitly at the **end of Section 1**:\
> *"This direct approach eliminates the need for vessel-specific training data or pre-trained vessel segmentation models, requiring only lightweight skull boundary extraction."*
>
> We have also expanded the Discussion (**Section 6, paragraph 2**) to quantify this overhead:\
> *"While our method requires skull segmentation from DSA, this is a lightweight task as cranial boundaries are high-contrast structures readily identifiable even in low-quality MAP images. Segmentation succeeded in all 77 cases, requiring minimal annotation effort (6 hours for 116 training images) with negligible inference overhead (0.1s per image)."*
>
> **(W2)** Methodologically, the objective mixes mNCC with a skull overlap term, the strongest gains appear to come from preprocessing + mask loss --- while the geodesic term provides modest average improvements and can degrade if $\lambda$ increases, suggesting sensitivity and limited robustness of the geometry prior.
>
> **(A2)** We agree that the primary performance gains stem from bilateral filtering and the cranium overlap loss. The geodesic consistency term serves a complementary role, providing additional guidance in challenging cases prone to local minima, as demonstrated in Figure 3. Regarding sensitivity to $\lambda$, we note that increasing $\lambda$ pushes the constraint toward strict orthogonality. Table 1 shows that enforcing exact 90° separation degrades performance, likely due to inherent deviations from perfect orthogonality in clinical acquisitions. Our soft formulation with $\lambda = 0.1$ balances geometric guidance with flexibility for real-world scanner configurations.
>
> To present this analysis more thoroughly, we have moved the ablation study from Appendix D to the main body (**Table 2**), with accompanying discussion in **Section 5, paragraph 3**:\
> *“An ablation on loss components is provided in Table 2. For geodesic consistency, $\lambda = 0.1$ was empirically found to perform well, as higher lambdas resulted in performance degradation. This finding aligns with our results on strictly orthogonal rotation, which also resulted in lower performance (see Table 1).”*
>
> **(W3)** The authors mentioned that head pose deviations can increase errors, yet no detailed analysis is further provided.
>
> **(A3)** We agree that such an analysis was missing from our original submission. In the updated manuscript, we have included an outlier analysis examining all cases with carotid artery $MCD_p$ > 10mm (N=8). These failures cluster into three categories: initialization errors due to non-standard head positioning (N=5), incomplete skull rendering from out-of-frame CTA acquisition (N=2), and insufficient structural information due to low DSA image quality (N=1). We now present representative examples of each failure mode in **Figure 4** and discuss these findings in **Section 5, paragraph 2**.\
> *“To characterize failure modes, we analyzed all cases where $MCD_p$ > 10mm for carotid arteries (N=8). These outliers fall into three categories: non-standard head positioning causing poor initialization (N=5), out-of-frame CTA acquisition resulting in incomplete skull rendering (N=2), and low DSA image quality yielding insufficient structural information (N=1). A representative example of each failure mode is shown in Figure 4.”*

---

> ### Author Response · Authors · 2026-01-23
> **Response to Reviewer nLRR (Part 2)**
>
> **Detailed comments:**
>
> **(DC1, Q1)** Add clarification on the “segmentation-free” claim: please rephrase to “no vessel segmentation required for optimization,” since skull segmentation is still required and is learned. Quantify how often skull segmentation fails and whether failures correlate with poor registration.
>
> **(A1)** We have reviewed all instances of "segmentation-free" throughout the manuscript to ensure we explicitly refer to vessel segmentation. Furthermore, we have added a clarification at the **end of Section 1**, now stating:\
> *"This direct approach eliminates the need for vessel-specific training data or pre-trained vessel segmentation models, requiring only lightweight skull boundary extraction."*
>
> Regarding skull segmentation reliability, the method succeeded in all 77 cases. For clarity, we have added a statement on this to **Section 6, paragraph 2**:\
> *"While our method requires skull segmentation from DSA, this is a lightweight task as cranial boundaries are high-contrast structures readily identifiable even in low-quality MAP images. Segmentation succeeded in all 77 cases, requiring minimal annotation effort (6 hours for 116 training images) with negligible inference overhead (0.1s per image)."*
>
> **Figure 4(c)** further illustrates robustness of cranium segmentation on a challenging example with heavy MAP artifacts, where the cranium is still correctly identified. Moreover, segmentation generalized to an external case from Radiopaedia without retraining (see **Appendix D**).
>
> **Questions to address in the rebuttal:**
>
> **(Q2)** How sensitive are results to initialization (pose priors) and to hyperparameters?
>
> **(A2)** Regarding initialization, our failure mode analysis (**Section 5, paragraph 2**) shows that non-standard head positioning - causing poor initialization - accounts for the majority of outlier cases (5 out of 8 cases with $\text{MCD}_{p}$​ > 10mm). An example of such a failure case is demonstrated in **Figure 4(a)** of the updated manuscript. Please see **(W3)** for a detailed response.
>
> Regarding hyperparameters, **Table 2** in the updated manuscript provides a systematic ablation. The method is most sensitive to $\lambda$: values that are too high push toward strict orthogonality and degrade performance (consistent with our results in Table 1), while $\lambda = 0.1$ provides a good balance. We refer to **(W2)** for an elaborate response.
>
> **(Q3)** Is there a practical solution to reduce runtime from around 4min to clinically viable latency (e.g., less than 1min) without requiring patient-specific pretraining?
>
> **(A3)** Upon profiling our optimization loop, we identified that most overhead stemmed from visualization callbacks executed at every iteration. Removing these reduces runtime to ~70 seconds on an NVIDIA L40S or 35 seconds on an NVIDIA H100, which we believe permits clinical deployment without patient-specific pretraining. We have updated **Section 5, paragraph 4** accordingly:\
> *“Total registration time was $\sim$70s per case (0.18 s/iteration) for both single and biplanar optimization on an NVIDIA L40S GPU, or 35s (0.09 s/iteration) on an NVIDIA H100 GPU.”*
>
> Additionally, we have extended our runtime discussion on optimization-based strategies to include optimization of the projection function through the use of CUDA kernel implementations (**Section 6, paragraph 4**):\
> *“Several optimization strategies could further reduce current runtime: sparse implementations of normalized cross-correlation would decrease computational overhead, CUDA kernel implementations of the forward projection could replace the current PyTorch-based approach, while improved hyperparameter selection could improve convergence rates.”*

---

### Official Review · Reviewer_njc7 · 2026-01-05

**Confidence:** 4
**Preliminary Rating:** 4

**Summary:**

paper proposes a segmentation-free framework for biplanar DSA/CTA registration. The method uses temporal MAP of DSA, refined with bilateral filtering and skull masking, to produce a silhouette representation compatible with CTA-derived DRRs. Registration is done through joint biplanar optimization of PA and lateral views, combining an image similarity loss with a geodesic consistency constraint that softly enforces near orthogonality between acquisition angles. The approach was validated evaluated and showed improvements compared to baseline optimization methods in terms several evaluation metrics.

**Strengths:**

The paper is well-structured and clearly explained. It presents a novel segmentation-free approach. By removing the need for vessel segmentationn, the method significantly reduces the data preparation burden. Also the integration of bilateral filtering, skull masking, and geodesic consistency provides a robust optimization framework capable of handling noisy DSA images and non-ideal biplanar geometries. Although currently optimization-based, the method establishes a foundation for learning based pose estimation with real-time inference.

**Weaknesses:**

These weaknesses primarily relate to robustness, scalability, and practical deployment, which are critical for clinical adoption. Addressing them would strengthen the manuscript's impact and reliability. (Detailed Comments)

**Detailed Comments:**

- The method is reported to be sensitive to nonstandard head positioning, which can limit its robustness in real-world applications. A systematic analysis of this issue, as well as potential mitigation strategies, is not fully addressed. The paper would benefit if the authors provided more details on general approaches that could be employed to address this limitation, or, if specifics are unavailable, outlined possible solution strategies.
- Current optimization takes ~4 minutes per case, which may be too slow for time-critical stroke interventions. are there any accelerated strategies or real-time feasibility avaialble for it ?
- Evaluation is limited to number of cases, which may not capture the full variability of patient anatomies or clinical scanner configurations. It raises questions about generalization across diverse clinical settings. Can the method be applied to other datasets ? expanding the analysis would significantly  improve the paper.
- While bilateral filtering, skull masking, and geodesic consistency are highlighted as important components, their individual contributions are not systematically analyzed. It would be valuable to discuss how removing or modifying each step affects the results and overall performance.

**Justification Of The Preliminary Rating:**

The paper addresses an interesting topic with a novel method that is clearly written and well-explained. Evaluation and validation are provided, and while there are minor issues, the work is generally sound and acceptable.

**Questions To Address In The Rebuttal:**

provided in detailed comments

---

> ### Author Response · Authors · 2026-01-23
> **Response to Reviewer njc7 (Part 1)**
>
> We thank the Reviewer for taking the time to provide a critical review of our work.
>
> **Detailed comments:**
>
> **(DC1)** The method is reported to be sensitive to nonstandard head positioning, which can limit its robustness in real-world applications. A systematic analysis of this issue, as well as potential mitigation strategies, is not fully addressed. The paper would benefit if the authors provided more details on general approaches that could be employed to address this limitation, or, if specifics are unavailable, outlined possible solution strategies.
>
> **(A1)** Thank you for this suggestion. In the revised manuscript, we have added an outlier analysis identifying all cases where $MCD_p$ > 10mm for carotid arteries (N=8). We categorize these into three failure modes: initialization errors from non-standard head positioning (N=5), poor optimization objectives due to low image quality (N=2), and out-of-frame anatomy (N=1). **Figure 4** in the updated manuscript illustrates an example of each failure mode. We have updated **Section 5, paragraph 2** to describe these results:\
> *“To characterize failure modes, we analyzed all cases where $MCD_p$ > 10mm for carotid arteries (N=8). These outliers fall into three categories: non-standard head positioning causing poor initialization (N=5), out-of-frame CTA acquisition resulting in incomplete skull rendering (N=2), and low DSA image quality yielding insufficient structural information (N=1). A representative example of each failure mode is shown in Figure 4.”*
>
> Regarding mitigation strategies, Section 6 discusses several directions for future work, including learned initialization to reduce sensitivity to starting pose. We have expanded this discussion to note that domain-adversarial training with gradient reversal could learn modality-invariant representations, enabling robust initialization across DSA and CTA despite their different intensity characteristics (**Section 6, paragraph 6**):\
> *“Domain-adversarial training with gradient reversal could further bridge the appearance gap between modalities, learning pose-discriminative features that are invariant to whether the input is a DSA projection or CTA-derived DRR.”*
>
> **(DC2)** Current optimization takes ~4 minutes per case, which may be too slow for time-critical stroke interventions. Are there any accelerated strategies or real-time feasibility available for it?
>
> **(A2)** We agree that the optimization time listed in our original manuscript would not allow for a time-sensitive scenario. Upon review of our optimization loop, we identified that most overhead stemmed from visualization callbacks executed at every iteration. Removing these reduces runtime to ~70 seconds on an NVIDIA L40S or 35 seconds on an NVIDIA H100, which would permit clinical deployment. We have updated **Section 5, paragraph 4** accordingly:\
> *“Total registration time was $\sim$70s per case (0.18 s/iteration) for both single and biplanar optimization on an NVIDIA L40S GPU, or 35s (0.09 s/iteration) on an NVIDIA H100 GPU.”*
>
> Further acceleration through learned initialization, as discussed in **Section 6, paragraph 6**, could enable real-time intraoperative use. Additionally, we have extended our discussion on iterative optimization strategies to include C++-based CUDA implementations of the projection function (**Section 6, paragraph 4**):\
> *“Several optimization strategies could further reduce current runtime: sparse implementations of normalized cross-correlation would decrease computational overhead, CUDA kernel implementations of the forward projection could replace the current PyTorch-based approach, while improved hyperparameter selection could improve convergence rates.”*

---

> ### Author Response · Authors · 2026-01-23
> **Response to Reviewer njc7 (Part 2)**
>
> **(DC3)** Evaluation is limited to number of cases, which may not capture the full variability of patient anatomies or clinical scanner configurations. It raises questions about generalization across diverse clinical settings. Can the method be applied to other datasets? Expanding the analysis would significantly improve the paper.
>
> **(A3)** We acknowledge that our evaluation cohort of 77 patients from a single institution is limited. To provide preliminary evidence of generalization, we applied our method to an external case obtained from Radiopaedia. As shown in the new **Appendix D (Figure 6)**, our method successfully registers this DSA and CTA volume despite differences in scanner configuration and acquisition protocol. While this single case cannot replace an extensive multi-center validation, it provides encouraging evidence that our approach is robust to domain shifts and can transfer between different scanner configurations and clinical settings.
>
> More broadly, our optimization-based approach may be more robust to domain shifts than learning-based alternatives, as we rely on intensity matching criteria (mNCC, dice overlap) that are agnostic to specific scanner configurations or acquisition protocols. Furthermore, since registration is performed pairwise, variations in patient anatomy are consistent across both modalities and can still be matched, unlike learning-based methods that must generalize across patients. We have added a discussion of this point in **Section 6, paragraph 4**:\
> *"While our evaluation is limited to single-institution data, the optimization-based nature of our approach may confer robustness to domain shifts. Intensity matching criteria generalize across scanner configurations, and pairwise registration inherently accommodates patient-specific anatomical variation. Preliminary evidence supporting this is provided by successful registration of an external case from Radiopaedia (see Appendix D). Nevertheless, comprehensive validation on multi-center data remains important future work."*
>
> **(DC4)** While bilateral filtering, skull masking, and geodesic consistency are highlighted as important components, their individual contributions are not systematically analyzed. It would be valuable to discuss how removing or modifying each step affects the results and overall performance.
>
> **(A4)** Such an analysis was indeed missing in the main body of our original submission. For our updated manuscript, we have moved the ablation study from Appendix D in the initial manuscript to Section 5 (Table 2 in the updated manuscript), providing a systematic analysis of the contribution of each component. An analysis of this ablation is included in **Section 5, paragraph 3**:\
> *"An ablation on loss components is provided in Table 2. For geodesic consistency, $\lambda = 0.1$ was empirically found to perform well, as higher lambdas resulted in performance degradation. This finding aligns with our results on strictly orthogonal rotation, which also resulted in lower performance (see Table 1). All other hyperparameters are as described in Table 3."*
>
> This analysis shows that bilateral filtering and the cranium overlap loss provide the primary performance gains, while the geodesic consistency term offers additional benefit in cases prone to local minima convergence.

---

### Official Review · Reviewer_ZAkC · 2026-01-08

**Confidence:** 5
**Preliminary Rating:** 4
**Final Rating:** 4

**Summary:**

This paper proposes GeoReg, a direct optimization-based method for registering biplanar DSA sequences to 3D CTA volumes without requiring vessel segmentation, using maximum-intensity projections (MAP) of DSA to recover a skull-based silhouette and differentiable DRR rendering from CTA. A geodesic consistency loss softly enforces approximate orthogonality between posteroanterior and lateral views, improving robustness over single-view or strictly constrained biplanar approaches. Evaluated on clinical stroke data, the method significantly improves vessel alignment metrics and demonstrates that reliable DSA–CTA registration is feasible without vessel-specific training data, simplifying clinical deployment

**Strengths:**

1 The manuscript is well written.
2 The proposed method is sound: The geodesic SO(3) consistency term exploits biplanar acquisition physics while remaining flexible to non-ideal, non-orthogonal clinical geometries—outperforming hard constraints
3 Results are great: Demonstrates consistent improvements in Dice, clDice, and centerline distance across carotid arteries and venous sinuses

**Weaknesses:**

1 To the best of my knowledge, vessel segmentation is a relatively mature (may be already solved) problem. Therefore, for a segmentation-free approach to be compelling in practice, it should either (1) demonstrate superior registration accuracy or (2) achieve a substantially reduced runtime compared to segmentation-based pipelines. The authors are encouraged to further discuss the practical advantages of the proposed method.

2 Figure 2 looks weird. the I_map_bm part

**Detailed Comments:**

Overall a solid work.

**Justification Of Final Rating:**

This paper proposes GeoReg, a direct optimization-based method for registering biplanar DSA sequences to 3D CTA volumes without requiring vessel segmentation, using maximum-intensity projections (MAP) of DSA to recover a skull-based silhouette and differentiable DRR rendering from CTA. A geodesic consistency loss softly enforces approximate orthogonality between posteroanterior and lateral views, improving robustness over single-view or strictly constrained biplanar approaches. Evaluated on clinical stroke data, the method significantly improves vessel alignment metrics and demonstrates that reliable DSA–CTA registration is feasible without vessel-specific training data, simplifying clinical deployment


The authors have addressed my concerns

**Justification Of The Preliminary Rating:**

This paper proposes GeoReg, a direct optimization-based method for registering biplanar DSA sequences to 3D CTA volumes without requiring vessel segmentation, using maximum-intensity projections (MAP) of DSA to recover a skull-based silhouette and differentiable DRR rendering from CTA. A geodesic consistency loss softly enforces approximate orthogonality between posteroanterior and lateral views, improving robustness over single-view or strictly constrained biplanar approaches. Evaluated on clinical stroke data, the method significantly improves vessel alignment metrics and demonstrates that reliable DSA–CTA registration is feasible without vessel-specific training data, simplifying clinical deployment


Method is OK, clinical value is slightly less OK.

**Questions To Address In The Rebuttal:**

The authors are encouraged to further discuss the practical advantages of the proposed method.

---

> ### Author Response · Authors · 2026-01-23
> **Response to Reviewer ZAkC**
>
> We thank the Reviewer for their kind feedback and suggestions.
>
> **Weaknesses and questions to address in the rebuttal**:
>
> **(W1, Q1)** To the best of my knowledge, vessel segmentation is a relatively mature (may be already solved) problem. Therefore, for a segmentation-free approach to be compelling in practice, it should either (1) demonstrate superior registration accuracy or (2) achieve a substantially reduced runtime compared to segmentation-based pipelines. The authors are encouraged to further discuss the practical advantages of the proposed method.
>
> **(A1)** Although we agree that single-modality vessel segmentation has matured, the bottleneck is obtaining matched segmentations across modalities with different resolutions, dimensionalities, contrast dynamics, and acquisition protocols. Furthermore, the superior temporal and spatial resolution of DSA yields more complete vascular trees than CTA, creating a correspondence problem, i.e. many vessels visible in DSA have no matching structure in CTA due to resolution limitations. This mismatch makes vessel-based registration ill-posed without extensive pruning heuristics to identify corresponding subsets. Additionally, reliable vessel correspondence typically relies on venous structures (as used in e.g. Downs et al.), yet venous phase acquisition is not universal in clinical DSA protocols, particularly in acute stroke settings where arterial-phase imaging is common. We now state this practical advantage more strongly in the **summation at the end of Section 1**:\
> *“[...], circumventing the cross-modal correspondence challenges inherent to vessel-based optimization.”*
>
> Furthermore, we discuss these practical advantages more extensively in the updated Discussion **(Section 6, paragraph 6)**:\
> *“Our results suggest that vessel-specific features may not be necessary, as skull boundaries alone provide strong geometric constraints for accurate registration. This offers a complementary approach that simplifies training requirements and broadens applicability to arterial-phase acquisitions, where venous structures are unavailable, and the superior resolution of DSA creates correspondence ambiguities with CTA vasculature.”*
>
> Secondly, we agree that the registration runtime of ~4 min reported in the manuscript would not permit clinical applicability. Upon profiling our optimization loop, we identified that most overhead stemmed from visualization callbacks executed at every iteration. Removing these reduces the runtime to ~70 seconds on an NVIDIA L40S or 35 seconds on an NVIDIA H100, allowing for clinically viable latency. We have updated the manuscript to reflect these corrected timings (**Section 5, paragraph 4**):\
> *“Total registration time was $\sim$70s per case (0.18 s/iteration) for both single and biplanar optimization on an NVIDIA L40S GPU, or 35s (0.09 s/iteration) on an NVIDIA H100 GPU.”*
>
> We provide additional insights on enhanced optimization strategies which would further substantially reduce runtime, now including the recently proposed CUDA-implemented projection functions, in **Section 6 (paragraph 4)**:\
> *“Several optimization strategies could further reduce current runtime: sparse implementations of normalized cross-correlation would decrease computational overhead, CUDA kernel implementations of the forward projection could replace the current PyTorch-based approach, while improved hyperparameter selection could improve convergence rates.”*
>
> **(W2)** Figure 2 looks weird. The $I_{MAP_{BM}}$ part.
>
> **(A2)** Thank you for pointing this out. We discovered that certain PDF renderers do not display the embedded images in **Figure 2** and Figure 4 (**Figure 5** in the updated manuscript) correctly, likely due to a compression artifact in our original export. To ensure consistent rendering across all PDF viewers, we have re-exported these figures as PNG files in the updated manuscript. We hope this resolves the issue. If the Reviewer's concern referred to a different aspect of the figure, we would greatly appreciate further clarification and will gladly address it during the discussion period.

---

### Author Rebuttal · Authors · 2026-01-23

**Rebuttal:**

We thank all reviewers for their constructive feedback. We have uploaded our revised manuscript with changes highlighted in red. All reviewers noted substantial runtime, which we were able to lower substantially by removing redundancies in the optimization loop. Two reviewers also noted that the paper would merit an analysis of outlier cases, which we now also provide in the updated manuscript.

We refer to the individual official comments for each reviewer for a detailed answer to each concern, and look forward to interacting during the discussion period.

**Supporting Material:**

/attachment/3f9032c99a69caa91fc8b0fa3f71e03a25809efc.pdf

---

### Meta-Review · Area_Chair_S5ri · 2026-02-04

**Recommendation:** Accept (Poster)
**Confidence:** 5

**Metareview:**

All reviewers were in consensus that the work should be accepted and I agree.

In my own abbreviated reading, I found the work to be largely building on the iterative optimization components of DiffDRR and DiffPose but with well-motivated regularizers and preprocessing steps. In the final version, please consider and integrate the following discussion points:
- The rebuttal claims that 35s for registration using an H100 is acceptable clinically. By no means is this close to deployability as real-time use requires near-interactive speeds.
- The 2nd paragraph of the preliminaries is significantly clearer technically than the rest of the paper leading up to that point. I highly suggest that it be also incorporated into the caption of fig. 1
- Future versions of this paper (e.g., for a journal) should incorporate experiments where the baselines are also trained with a segmentation loss on the skull, as in this work.

---

### Decision · Program_Chairs · 2026-02-13

Accept (Poster)